# Direct Production of Ni–Co–Mn Mixtures for Cathode Precursors from Cobalt-Rich Lithium-Ion Battery Leachates by Solvent Extraction

**Niklas Jantunen** [1,*] , **Sami Virolainen** [1] and **Tuomo Sainio** [2]

1   School of Engineering Science, LUT University, Yliopistonkatu 34, 53850 Lappeenranta, Finland
2   School of Engineering Science, LUT University, Mukkulankatu 19, 15210 Lahti, Finland
*   Correspondence: niklas.jantunen@lut.fi

**Abstract:** A novel solvent extraction scheme was developed for the processing of Co-rich lithium-ion battery (LIB) leachate to a Ni–Co–Mn (NCM) sulfate mixture that can be directly used in the precursor synthesis of LIB cathodes. Conventional hydrometallurgical recycling of spent LIBs usually aims at separation of Li, Ni, Co, and Mn into pure fractions, which is simplified here. Operating pH and the number of extraction stages for each separation were evaluated from batch equilibrium experiments. Two continuous countercurrent extractions with bis(2-ethylhexyl) hydrogen phosphate (D2EHPA) and one with Cyanex 272 were studied in bench-scale mixer-settler equipment, and a Ni–Co–Mn solution with $n$(Ni):$n$(Co) = 14.16 and $n$(Ni):$n$(Mn) = 8.06 was obtained. The Ni:Co:Mn molar ratio in the NCM mixture can be adjusted to, for example, 8:1:1 using a Co-rich raffinate from the same process, and no additional transition metal salts are required for tuning the composition. Stripping raffinate containing 102.7 g L$^{-1}$ Co at 99.8% relative purity was obtained from Cyanex 272 extraction. The main benefit of the process concept is that the solvent extraction separations can be operated with less stringent requirements than when producing pure metal salts.

**Keywords:** continuous solvent extraction; flowsheet development; battery recycling; lithium; nickel; manganese; cobalt

## 1. Introduction

Recycling of lithium-ion batteries (LIBs) has recently gained a lot of attention due to rapidly increasing use of battery-powered devices and vehicles. The larger demand for LIBs will not only increase the consumption of raw materials but the growth in LIB sales will also result in an increasing number of end-of-life batteries in the future. Sustainability imposes recovery of the spent raw materials, especially metals. Chagnes and Pospiech (2013) [1], Li et al. (2018) [2], and Meshram et al. (2014) [3] have published extensive reviews on recycling of spent LIBs. Metals are recovered from the spent LIBs using combinations of mechanical, pyrometallurgical, and hydrometallurgical processes.

The hydrometallurgical recycling of LIBs involves the liberation of metals from the spent cathode powders by reductive acid leaching. Many different mineral- and carboxylic acids can be used for leaching [1–3] but $H_2SO_4$ with $H_2O_2$ as a reductant can be recommended due to high leaching recoveries and economical advantage [4]. Hence, the discussion in this paper is focused on sulfate solutions. The LIB leachates contain the key cathode metals (Li, Co, Ni, and Mn), and the presence of Al, Fe, and Cu in small quantities is common [2,3]. Since the metal concentrations in the leachates vary depending on the composition of the utilized battery waste, the heterogeneous solutions require further processing before the recovery of the valuable metal fractions as products or intermediates. Solvent extraction, precipitation and/or ion exchange are well-known methods for the separation and purification of metals from battery leachates [1–3,5]. Especially solvent



extraction is advantageous, because the equilibrium chemistry and kinetics with commercial extractants enable the processing of large volumes of solutions at a low temperature (e.g., 20–50 °C) in a relatively short time. Furthermore, high recoveries and separation efficiencies can be achieved.

Solvent extraction processes often aim at the isolation of the metals into their own fractions from where they can be precipitated or crystallized as pure metal salts or electrowon [4–11]. However, due to the growing battery markets and shift towards Ni-rich NCM cathodes in LIB manufacturing, a multi-metal solution with specific amounts of Ni, Co, and Mn (NCM-solution) can also be one product of the solvent extraction process [12–14]. Recently, Liu et al. (2021) [12] used bis-(2-ethylhexyl)phosphinic acid (P227) to co-extract Ni, Co, and Mn from chloride leachates of NCM622 and NCM811 cathode powders. The loaded Ni, Co, and Mn were stripped and co-precipitated as oxalates, which were further processed to NCM111 and NCM622 cathode materials. A similar co-extraction approach has been studied in sulfate media by Shuya et al. (2020) [13] and Yang et al. (2017) [14]. Shuya et al. (2020) [13] used Versatic 10 (neodecanoic acid) for the co-extraction of Ni, Co, and Mn from Ni-rich sulfate leachate, whereas Yang et al. (2017) [14] used bis(2-ethylhexyl) hydrogen phosphate (D2EHPA) for NCM111 leachate.

On the other hand, the LIB leachates may be rich in Co [15,16], or otherwise not matching any commercial NCM cathode composition. In these cases, the composition of the co-precipitation synthesis mixture must be adjusted using additional transition metal salts, as Yang et al. (2017) [14] have suggested, or the excess metal content must be separated from the leachate. This paper presents a solvent extraction flowsheet that separates most of the Li, Co, and Mn from a Co-rich LIB leachate and produces a Ni-rich NCM synthesis mixture with desired composition as another product. Here, the compositional tuning and pre-concentration of the NCM synthesis mixture is done by solvent extraction alone, without using additional transition metal salts.

The processing of the LIB leachates begins with the removal of impurities (here Al, Fe, and Cu), which can be accomplished by NaOH precipitation [17], ion exchange [18], or solvent extraction [19]. The extractive method is recommended because of the potential loss of Co during the hydroxide precipitation [8,20] and part of the Mn may be lost together with the impurity metals during ion exchange [18]. Once the impurities are removed, Ni, Co, and Mn are extracted to the organic phase by suitable extractants, while at least most of the Li remains in the raffinate. Acidic organophosphorus extractants D2EHPA, Cyanex 272 (bis(2,4,4-trimethylpentyl)phosphinic acid), and PC88-A (2-ethylhexoxy(2-ethylhexyl)phosphinic acid) are well-known extractants for this kind of metal separation. Mn can be separated from Li, Co, and Ni using D2EHPA, but the separation of Co and Ni by D2EHPA is difficult [21,22]. On the other hand, Cyanex 272 has excellent Co/Ni selectivity but weak Mn/Co selectivity [22]. The Co/Ni selectivity of PC88-A is not as good as Cyanex 272 [23], and PC88-A does not exhibit phenomenal Mn/Co selectivity, either [11,24]. Instead of using a single extractant, Chen and Ho (2018) [4] suggested the separation of Li, Ni, Co, and Mn by separate Cyanex 272 and D2EHPA circuits. Mn and Co were separated from Li and Ni by Cyanex 272, and Ni was selectively precipitated from the raffinate (containing Li and Ni) using dimethylglyoxime. Mn and Co were separated by D2EHPA from the mixture of $MnSO_4$ and $CoSO_4$, which was obtained by stripping of the loaded Cyanex 272. Weak extraction of Li by D2EHPA, Cyanex 272, and PC88-A has been typically reported (<10%) at pH ≤ 6 [4–6,10,20,24], but the extraction of Li becomes significant when there are plenty of free extractant ligands [14,25]. D2EHPA and Cyanex 272 were selected as the extractants for this work because it was recognized that the overall material efficiency could be improved with the industry-proven extractants by an improved flowsheet. The key extraction steps of the flowsheet were studied by batch equilibrium experiments, and the numbers of required extraction stages for countercurrent extractions were evaluated by the McCabe–Thiele method. The key steps within the presented process concept were experimentally validated by continuous countercurrent experiments in mixer-settlers.

## 2. Materials and Methods

### 2.1. Chemicals

A synthetic aqueous solution (Table 1) was prepared to simulate a purified LIB leachate by dissolving appropriate amounts of $Li_2SO_4 \cdot H_2O$ ($\geq$99%, Acros Organics, Thermo Fisher (Kandel) GmbH, Kandel, Germany), $NiSO_4 \cdot 6H_2O$ (99%, GPR RECTAPUR®, VWR Chemicals, VWR International, Leuven, Belgium), $CoSO_4 \cdot 7H_2O$ ($\geq$97%, VWR Chemicals, VWR International, Leuven, Belgium), $MnSO_4 \cdot H_2O$ (99.8%, AnalaR NORMAPUR® ACS, Reag. Ph. Eur., VWR Chemicals, VWR International, Leuven, Belgium), and $H_2SO_4$ (95–97%, GPR RECTAPUR®, VWR Chemicals, VWR International, Leuven, Belgium) in pure water (de-ionized and purified by reverse osmosis). Redox potential of the solution was +590 $\pm$ 15 mV vs. SHE, and its pH was 1.0. Al, Fe, and Cu were not included here in the simulated leachate because effective methods for their removal have been reported earlier [17–19], and the removal of impurities was not in the scope of the current work. $Na_2SO_4$ (99.5%, AnalaR NORMAPUR® ACS, Reag. Ph. Eur., VWR Chemicals, VWR International, Leuven, Belgium) was used for ionic strength adjustment when metal solutions with experimentally determined raffinate compositions were prepared for subsequent extraction experiments.

**Table 1.** Composition of the simulated LIB leachate. Metal concentrations adopted from [26].

|  | Li | Ni | Co | Mn | H$_2$SO$_4$ | *I* (Ionic Strength) |
|---|---|---|---|---|---|---|
| $c$ [g L$^{-1}$] | 2.50 | 2.00 | 16.8 | 2.10 | 15.0 | – |
| $c$ [mmol L$^{-1}$] | 360 | 34.1 | 285 | 38.2 | 306 | 2430 |

0.8 M D2EHPA solution with 5 vol-% TBP as a phase modifier was prepared by dissolving bis(2-ethylhexyl) hydrogen phosphate (97%, Merck KGaA, Darmstadt, Germany) and tri-*n*-butyl phosphate (98%, Alfa Aesar, Thermo Fisher (Kandel) GmbH, Germany) in Exxsol D80 (ExxonMobil, Irving, TX, USA) aliphatic hydrocarbon mixture. Third phase formation will occur with D2EHPA and Exxsol D80 under heavy Na loading if a phase modifier is not used [27]. Cyanex 272 (85.3%, bis(2,4,4-trimethylpentyl)phosphinic acid, Solvay, Brussels, Belgium) was dissolved in Exxsol D80 without a separate phase modifier because Cyanex 272 contains 10–15% tris(2,4,4-trimethylpentyl)phosphine oxide. The Cyanex 272 solution was prepared at 0.8 M concentration. D2EHPA, Cyanex 272, TBP, and Exxsol D80 were used as received without pre-conditioning, except when stripping of metals from the loaded extractants was studied.

### 2.2. Batch Experiments

Batch extractions were carried out in a 1.0 dm³ jacketed glass reactor. The pH was adjusted using aqueous NaOH solutions. Temperature was controlled by an external thermostat. O/A = 1 was used in determination of the pH isotherms, unless specifically stated otherwise. Loading isotherms were determined in a similar manner but the O/A ratio was varied, whereas pH was maintained constant. Back-extraction performance was studied by stripping the loaded extractants with $H_2SO_4$ at various O/A ratios without other pH adjustment. Equilibrium time was 15 min.

### 2.3. Continuous Experiments

MEAB MSU0,5 mixer-settler units (MEAB Metallextraktion AB, Askim, Sweden) made of PTFE and PFA were used in the continuous extraction experiments. The mixer volume of a single MSU0,5 unit is 120 cm³ and the volume of the settler is 460 cm³. The horizontal cross-section area of the settler is 55 cm². The aqueous sulfate solutions and the extractant solutions were fed by PTFE diaphragm pumps (ProMinent DLTA), and a dual-channel syringe pump (Gemini 88) was used for injecting NaOH solution to control pH. Mettler-Toledo InLab® Semi-Micro-L electrodes and Consort C3060 were used for measuring and logging the pH values. All continuous experiments were done at room

temperature ($21 \pm 1$ °C) and a minimum of five cascade volumes of liquid was processed in each experiment, as suggested by the conventional well-known theory on residence time distributions.

### 2.4. Analytical Methods

The samples were centrifuged to minimize entrainments before further analytical procedures. Metal concentrations were determined from both phases using ICP-MS (Agilent 7900). D2EHPA extract samples were backextracted at A/O = 20 with 5 N HCl, and the HCl raffinates from backextraction were analyzed. Similar backextraction analysis was done for the Cyanex 272 extracts, but 3 M $H_2SO_4$ was used instead of HCl for stripping of the metals.

### 2.5. Data Treatment

The percentage of extraction was calculated using Equation (1)

$$E = \frac{100D}{D + \frac{V_{aq}}{V_{org}}} \tag{1}$$

where $E$ is the percentage of extraction [%], $D$ is the distribution ratio ($D = [M]_{org}/[M]_{aq}$ where $M$ denotes a metal), $V_{aq}$ is the volume of the aqueous phase, and $V_{org}$ is the volume of the organic phase.

The relative purity was calculated using Equation (2)

$$P_R(M) = 100\% \cdot \frac{w_M}{\sum_i w_i} \tag{2}$$

where $P_R(M)$ is the relative purity of metal $M$ [wt.%], $w$ is weight concentration [g L$^{-1}$], and the summation in the denominator includes all metals in the solution.

### 3. Results and Discussion

A solvent extraction process (Figure 1) for producing $Li_2SO_4$, $CoSO_4$, $MnSO_4$, and NCM synthesis mixtures from LIB leachates was conceptualized based on the existing knowledge on solvent extraction (see Introduction) and the requirements set by hydroxide co-precipitation synthesis of NCM cathode precursors.

The hydroxide co-precipitation syntheses are often carried out from solutions that contain Ni, Co, and Mn at a specific molar ratio (e.g., 8:1:1) at 2 mol L$^{-1}$ total concentration of Ni, Co, and Mn [28–31]. Here, the concentrated NCM sulfate mixture is obtained via solvent extraction by stripping a metal-bearing extractant solution with moderately concentrated acid at a high solvent-to-feed (S/F) ratio. To produce an NCM sulfate mixture for the co-precipitation synthesis of, e.g., NCM811, Ni:Co, and Ni:Mn, molar ratios in the organic phase must be 8. If one or both of the ratios are higher than 8, the composition in the final NCM mixture is adjusted to the desired stoichiometric ratio using the concentrated $CoSO_4$ and $MnSO_4$ products from the same process. Here (Figure 1), a loaded D2EHPA solution with suitable amounts of Ni, Co, and Mn (NCM-D2EHPA) is obtained by mixing the NiSX extract with an extract bleed from MnSX. Alternatively, the partially loaded D2EHPA from the MnSX circuit could be used for the extraction of Ni. Co can be delivered to the NCM-D2EHPA within the NiSX extract and/or within the MnSX extract bleed because of the co-extraction of Co in both MnSX and NiSX. Therefore, a small amount of Co can be allowed in the CoSX raffinate (Figure 1) and one extraction stage is likely sufficient for the partial separation of Co. Design of the extraction and stripping stages in the process scheme was studied experimentally and will be discussed in the following sections.

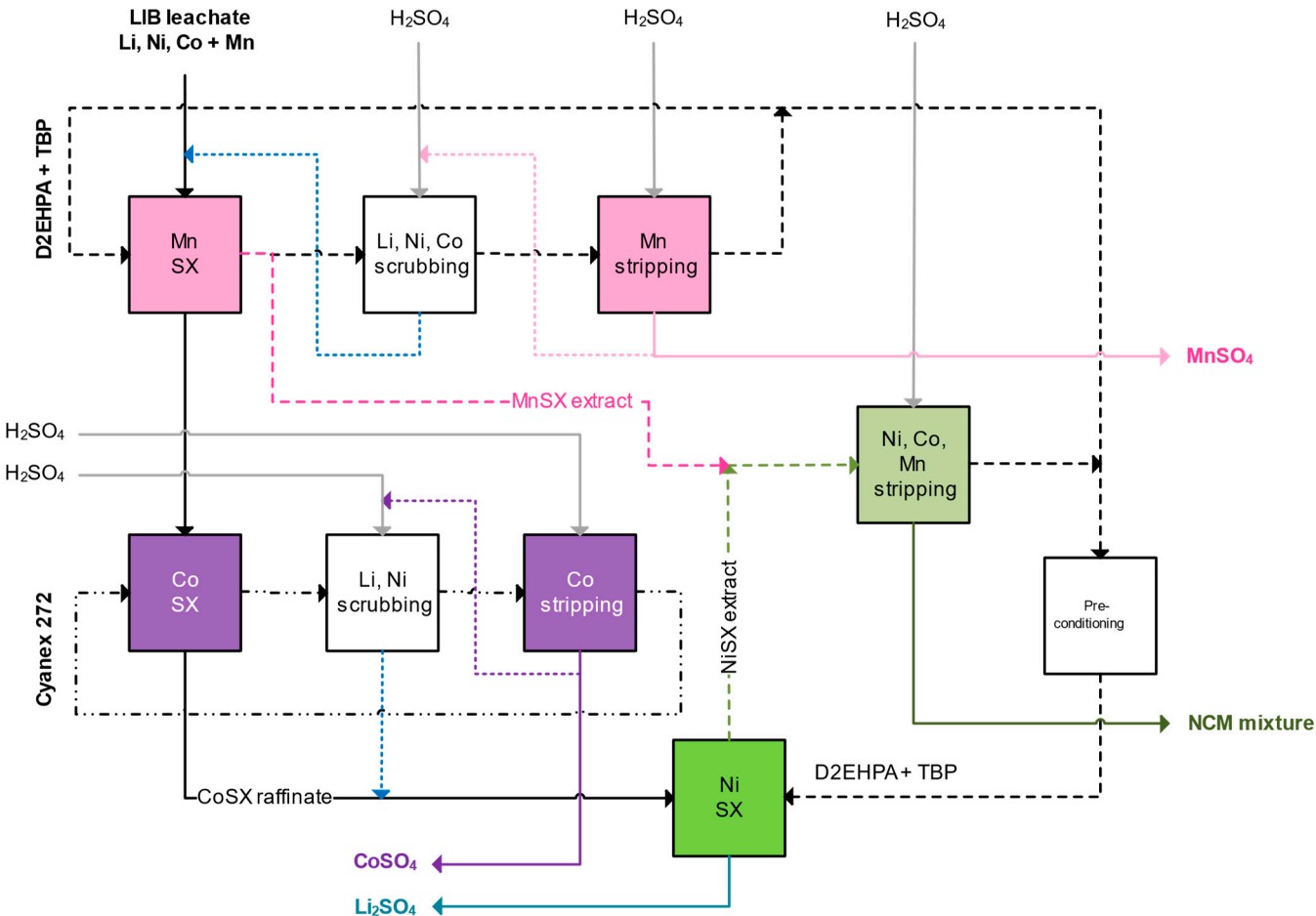

**Figure 1.** A simplified diagram of a solvent extraction process for the processing of lithium-ion battery leachates. The blocks studied in this work are colored. Dashed line = D2EHPA stream (organic), dashdot line = Cyanex 272 stream (organic), solid line = aqueous stream, dotted line = bleed-off from stripping raffinate (aqueous).

### 3.1. Selecting the pH for Mn, Co, and Ni Extraction Stages

Mn is the most easily extracted metal by D2EHPA among the metals in the simulated leachate (Table 1), and over 95% Mn was extracted from the simulated leachate by 0.8 M D2EHPA at pH $\geq$ 3.14 (Figure 2a). The $pH_{50}$ values in extraction by 0.8 M D2EHPA were approximately 1.98 for Mn, 3.95 for Co, and 4.70 for Ni as determined by graphical interpolation. These $pH_{50}$ values agree reasonably well with those given by Sole (2018) [22] and further emphasize why the Co–Ni separation by D2EHPA is more challenging than Mn–Co and Mn–Ni separations. $Na^+$ ions were introduced into the Li–Ni–Co–Mn sulfate system due to pH adjustment (Figure 2a), hence the extraction of Na. The extraction of Co and Ni was incomplete likely because of the extractant saturation. According to the ICP-MS analysis, the loaded 0.8 M D2EHPA contained 0.11 mol $L^{-1}$ Li, 0.18 mol $L^{-1}$ Na, 0.03 mol $L^{-1}$ Ni, 0.26 mol $L^{-1}$ Co, and 0.04 mol $L^{-1}$ Mn in equilibrium at pH = 7.7. These concentrations seem to suggest over-stoichiometric extraction when assuming 2:1 ligand-to-metal stoichiometry for the extraction of Ni, Co, and Mn; and 1:1 extraction stoichiometry for Li and Na, respectively. This may be explained by surfactant effects, such as formation of reversed micelles or microemulsions [27].

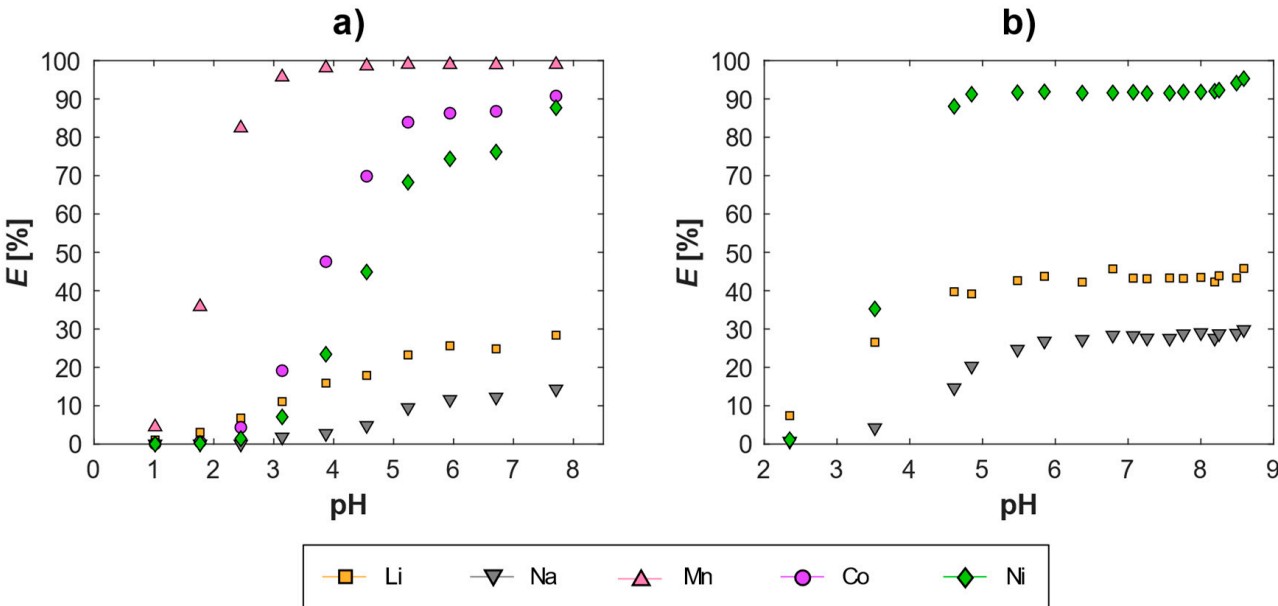

**Figure 2.** The effect of pH on the extraction of Li, Ni, Co, Mn, and Na by 0.8 M D2EHPA (modified by 5 vol-% TBP and diluted in Exxsol D80). $T = 25 \pm 1$ °C; $t_{eq}$ = 15 min; O/A = 1. Initial aqueous concentrations [g L$^{-1}$]: (**a**) Li: 2.52; Ni: 2.05; Co: 16.5; Mn: 2.09; (**b**) Li: 2.29; Na: 26.2; Ni: 1.94.

To study the effect of pH on extraction of Ni and Li by 0.8 M D2EHPA, an aqueous solution containing Li$_2$SO$_4$, NiSO$_4$, and Na$_2$SO$_4$ was used in extraction experiments to simulate Mn- and Co-free raffinate. Instead of H$_2$SO$_4$, Na$_2$SO$_4$ was used to adjust the ionic strength to a similar level with the simulated leachate (Table 1) because Na$^+$ will be introduced into the system if NaOH is used for pH adjustment during the extraction of Mn and Co (Figure 1). A total of 91.2% Ni was extracted at pH = 4.85 but further increases in pH had a limited effect in increasing the extraction of Ni and Li (Figure 2b). The pH isotherms of Ni and Li were shifted toward lower pH values in the absence of Mn and Co (Figure 2), and much higher amounts of Ni, Li, and Na were extracted from the Li$_2$SO$_4$–NiSO$_4$–Na$_2$SO$_4$ system than the simulated leachate. Increases in pH above 4.85 mostly increased the extraction of Na, and an increase in [Na]$_{org}$ from 8.0 to 13.3 g L$^{-1}$ was observed due to addition of NaOH. At pH $\geq$ 4.85 there was 1.6–1.7 g L$^{-1}$ Ni and 0.9–1.1 g L$^{-1}$ Li in the organic phase, meaning that most of the extractant was in the form of Na-D2EHPA.

Cyanex 272 requires a higher pH than D2EHPA for extracting the metals since bis(2,4,4-trimethylpentyl)phosphinic acid is a weaker acid than D2EHPA. The pH$_{50}$ values in extraction by 0.8 M Cyanex 272 were 3.82 for Mn and 4.44 for Co. The pH$_{50}$ value (Figure 3a) for Co is in close agreement with the pH isotherm reported by Virolainen et al. (2017) [10] for 1 M Cyanex 272, and the minor difference is explained by different extractant concentration. Extraction of Li, Ni, or Na by 0.8 M Cyanex 272 was minimal at O/A = 1 (Figure 3a). However, the extraction of Co was incomplete (96.6%) at O/A = 1 and pH = 6.35 because there was more than enough Mn and Co to load the extractant near its practical maximum capacity. Cyanex 272 turns into an extremely viscous gel under heavy Co loading, and the rheology would not allow operation above pH = 6.35 at O/A = 1. The viscosity increase is related to polymerization of the Co-loaded Cyanex 272 when the loading of Co reaches 70–75% of the theoretical extractant capacity [32]. Since Cyanex 272 cannot be used in practice under high Co loading, the pH isotherms for 0.8 M Cyanex 272 were determined also using O/A = 2.5 (Figure 3b). An increase in the amount of extractant enables a higher amount of metals to be extracted and shifts the pH isotherms towards lower pH values (Figure 3b). In total, 99.8% Mn and 99.0% Co were extracted by Cyanex 272 at pH = 5.27 and O/A = 2.5, whereas 13.9% Ni and 3.6% Li were co-extracted (Figure 3b). Both Cyanex

272 and D2EHPA extract Li and Na when there are not enough more extractable cations (such as $H^+$ and $Mn^{2+}$) to occupy the extractant ligands (Figures 2 and 3). This observation is consistent with [14,25] and similar behavior has also been reported with Versatic 10 [13].

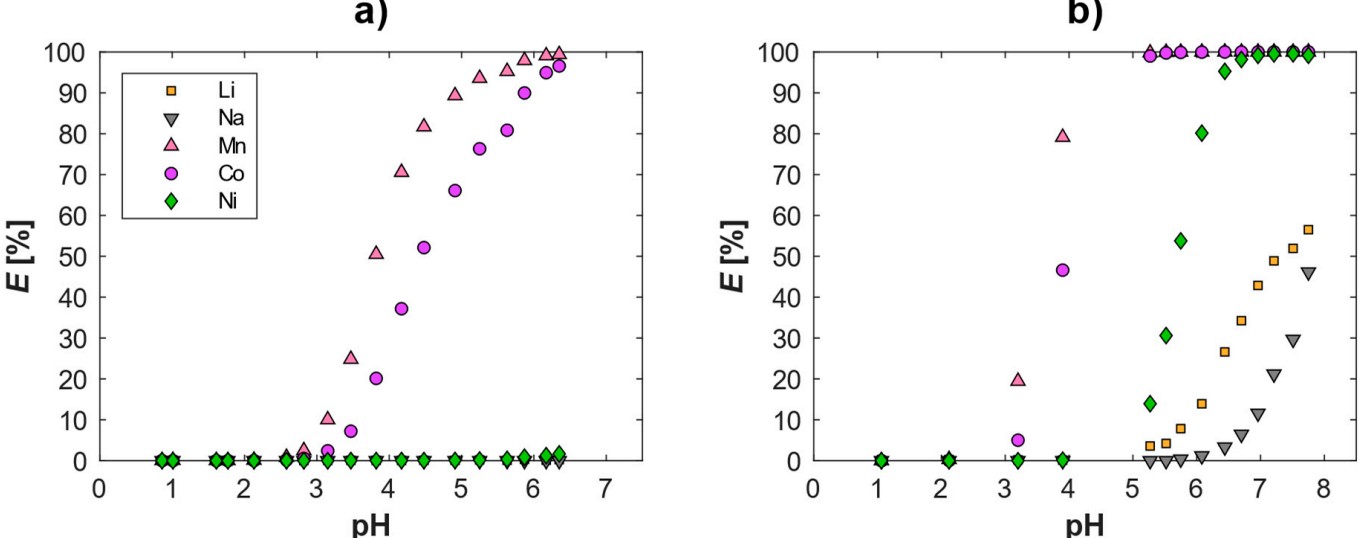

**Figure 3.** The effect of pH on the extraction of Li, Ni, Co, Mn, and Na by 0.8 M Cyanex 272 (diluted in Exxsol D80). $T = 25 \pm 1\,°C$; $t_{eq} = 15$ min. Initial aqueous concentrations [g L$^{-1}$]: Li: 2.52; Ni: 2.05; Co: 16.5; Mn: 2.09. Subfigures: (**a**) O/A = 1; (**b**) O/A = 2.5.

*3.2. Separation of Mn from Li, Ni, and Co by 0.8 M D2EHPA*

All the Mn was extracted by 0.8 M D2EHPA, but co-extraction of the other metals increased as the number of metal-free extractant ligands increased (Figures 2a and 4). The relative purity of Mn in the organic phase is increased when O/A is lowered, but a high recovery of Mn cannot be obtained by single extraction using small O/A (Figure 4b). Therefore, multiple extraction stages will be required to achieve both a high percentage of extraction and a good selectivity for Mn. McCabe–Thiele analysis suggested that 99.5% of the Mn could be removed by 0.8 M D2EHPA in three ideal countercurrent extraction stages operating with S/F = 0.8 at pH = 2.50 (Figure 5). A total of 94.2% Mn was extracted from the simulated leachate in three countercurrent stages (Table 2) and the raffinate contained 0.12 g L$^{-1}$ Mn. The experimental results (Table 2) are different from the theoretical prediction (Figure 5) likely because the raffinate exit stage operated at pH = 2.08, whereas the prediction assumes pH = 2.50 for all stages. Moreover, chemical equilibrium is often not achieved in continuous extractors, i.e., the stage efficiency is below 100%. The separation of Mn from Li, Ni, Co, and Na can be further improved by lowering S/F and increasing the number of extraction stages. Considering the process concept in Figure 1, complete removal of Mn is more important than minimizing the co-extraction of Li, Ni, and Co. Any Mn remaining in the raffinate will be extracted in the next process step by Cyanex 272 during the extraction of Co and will negatively impact the purity of the CoSO$_4$ product (Figure 1, see Section 3.3).

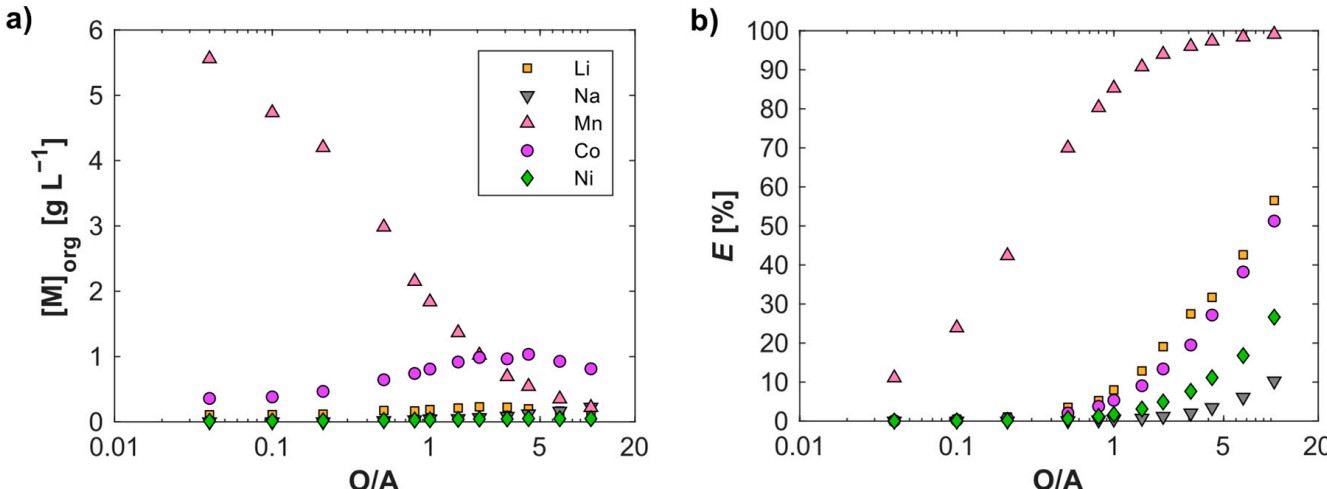

**Figure 4.** Effect of volumetric phase ratio on (**a**) metal concentrations in the organic phase and (**b**) the percentage of Li, Ni, Co, Mn, and Na extracted by 0.8 M D2EHPA (modified by 5 vol-% TBP and diluted in Exxsol D80). $T = 25 \pm 1$ °C; pH = 2.5; $t_{eq}$ = 15 min. Initial aqueous concentrations [g L$^{-1}$]: Li: 2.63; Ni: 1.98; Co: 16.2; Mn: 2.07. Note the logarithmic O/A-axis.

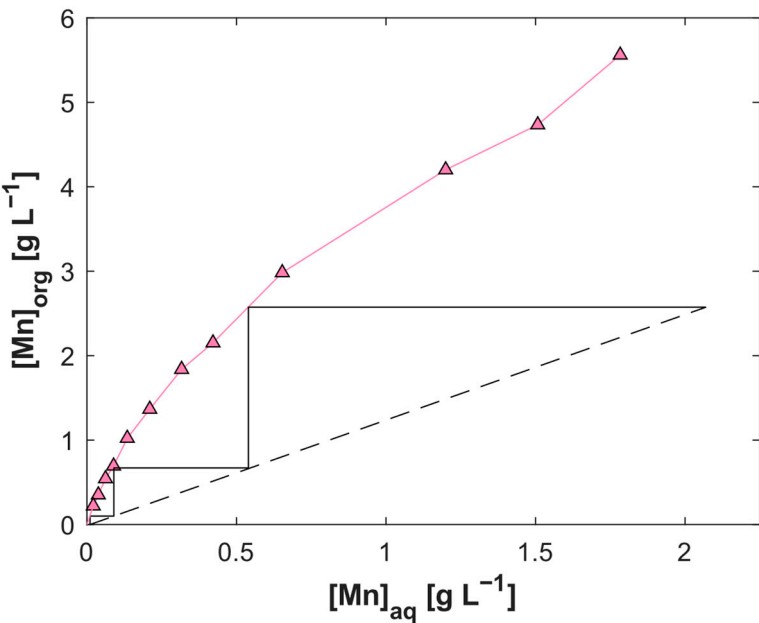

**Figure 5.** A McCabe–Thiele analysis applied on the equilibrium distribution of Mn between 0.8 M D2EHPA (modified by 5 vol-% TBP and diluted in Exxsol D80) and simulated LIB leachate. $T = 25 \pm 1$ °C; pH = 2.5; $t_{eq}$ = 15 min. Initial aqueous concentrations [g L$^{-1}$]: Li: 2.63; Ni: 1.98; Co: 16.2; Mn: 2.07. The operating line was calculated with S/F = 0.8 and 99.5% Mn removal.

**Table 2.** Compositions of the extract and raffinate in three-stage countercurrent extraction of Mn from a simulated LIB leachate by 0.8 M D2EHPA. S/F = 0.8; $T = 21 \pm 1$ °C; $\tau_{mix}$ = 4 min. Concentrations in [mg L$^{-1}$]. Numbering of the stages is started from the organic feed stage.

| | **[Li]** | **[Na]** | **[Mn]** | **[Co]** | **[Ni]** | **pH$_{avg}$** |
|---|---|---|---|---|---|---|
| Simulated leachate | 2340 | 6930 | 2070 | 16,200 | 1820 | Stage 1: 2.08 ± 0.03 |
| Raffinate | 2180 | 8300 | 115 | 15,700 | 1790 | Stage 2: 2.52 ± 0.06 |
| Extract | 151 | 21.2 | 2970 | 541 | 17.7 | Stage 3: 2.49 ± 0.02 |

Co, Ni, and Li were co-extracted during the extraction of Mn from the simulated leachate (Table 2). Small amounts of co-extracted Li, Ni, and Co in D2EHPA (Figure 4; Table 2) are not a concern considering the production of NCM mixtures but scrubbing of the Mn-rich extract with dilute $H_2SO_4$ and/or $MnSO_4$ solution is recommended so that a purified $MnSO_4$ solution can be obtained (Figure 6). Li, Ni, Co, and Mn will be completely stripped when the amount of $H_2SO_4$ is sufficiently high (Figure 6b,d), but the separation of Mn is enhanced when the amount of $H_2SO_4$ is limited with respect to the stoichiometric amount of Mn. In total, 41.3% Li, 96.2% Ni, 79.9% Co, and 3.9% Mn were backextracted from the loaded 0.8 M D2EHPA by 0.5 M $H_2SO_4$ at O/A = 50 (Figure 6b). A concentrated $MnSO_4$ solution can be obtained by stripping the Mn-D2EHPA with moderately concentrated $H_2SO_4$ (Figure 6c) at large O/A ratio, and a bleed from the concentrated $MnSO_4$ raffinate can be used for scrubbing (Figure 1). Then, the raffinate from scrubbing can be blended in the LIB leachate, which enters the Mn extraction stages (Figure 1).

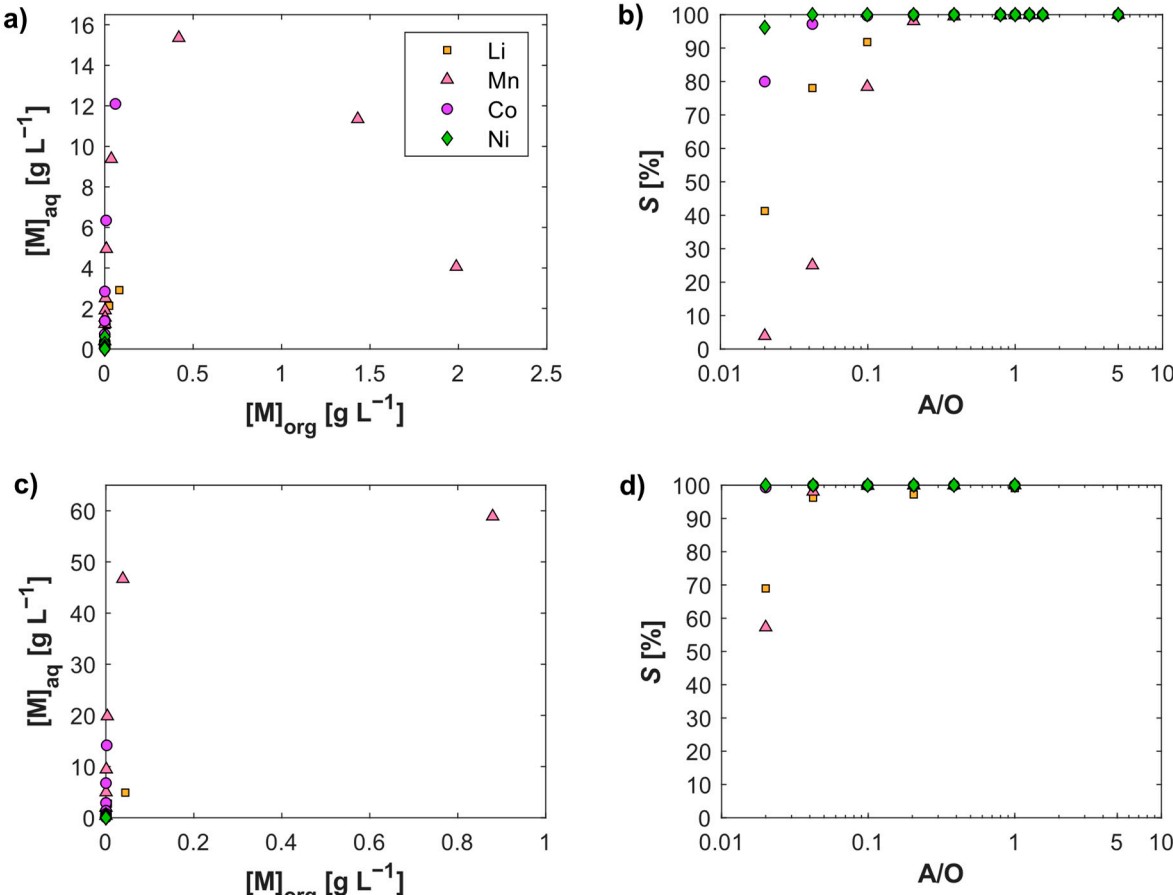

**Figure 6.** Backextraction of Li, Ni, Co, and Mn by $H_2SO_4$ from loaded 0.8 M D2EHPA (modified by 5 vol-% TBP and diluted in Exxsol D80). $T = 25 \pm 1\,°C$; $t_{eq} = 15$ min. Initial concentrations in the organic phase [g $L^{-1}$]: Li: 0.15; Ni: 0.01; Co: 0.3; Mn: 2.02. Subfigures: (**a**) Distribution of metals in stripping with 0.5 M $H_2SO_4$; (**b**) Percentages of metals stripped with 0.5 M $H_2SO_4$; (**c**) Distribution of metals in stripping with 2 M $H_2SO_4$; (**d**) Percentages of metals stripped with 2 M $H_2SO_4$. Note the logarithmic A/O-axis in subfigures (**b**,**d**).

### 3.3. Separation of Co from Li and Ni by Cyanex 272

Since Mn can be completely extracted from the simulated leachate, the loading isotherm of Co with 0.8 M Cyanex 272 (Figure 7) was determined for a Mn-free solution at pH = 5.3, which was chosen based on the pH isotherms (Figure 3b). The highest $[Co]_{org}$ (14.5 g $L^{-1}$) (Figure 7a) corresponds to approximately 61.5% of the theoretical loading capacity of 0.8 M Cyanex 272. However, 0.8 M Cyanex 272 was highly selective for

Co at pH = 5.3 with O/A ratios between 0.04–1.5 (Figure 7c), meaning that $[Co]_{org}$ can be maintained somewhat lower than at 14 g $L^{-1}$ to avoid problems related to high viscosity (see chapter 3.1). Although the concentrations of Li and Ni were low (<0.3 g $L^{-1}$) in the Cyanex 272 extracts (Figure 7b), their extraction became significant after the aqueous phase was practically free from Co (Figure 7c).

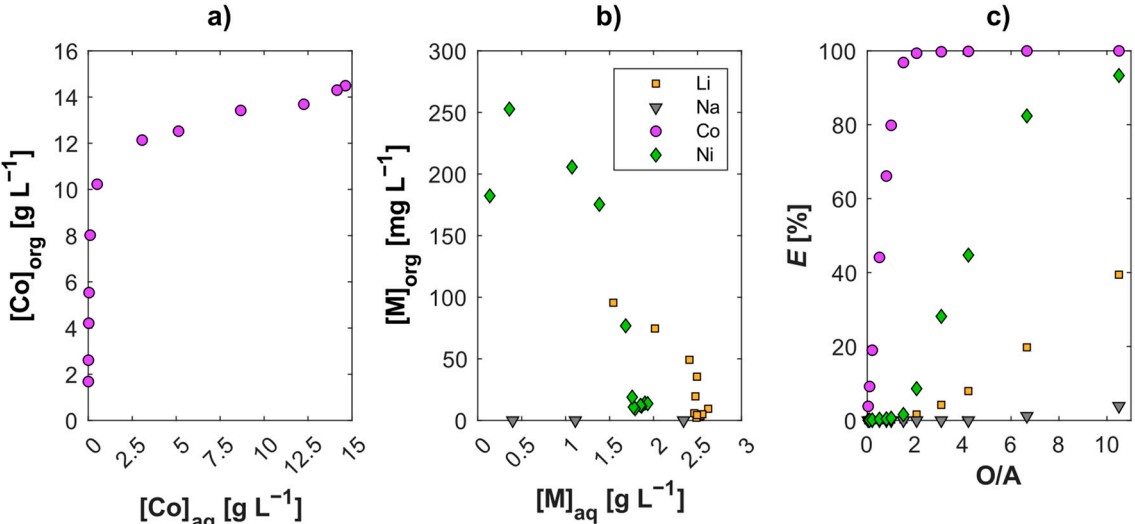

**Figure 7.** Distribution of Li, Ni, Co, and Na between 0.8 M Cyanex 272 (diluted in Exxsol D80) and Mn-free simulated LIB leachate: (**a**) Concentrations of cobalt in equilibrium; (**b**) Concentrations of Li, Ni, and Na in equilibrium; (**c**) Effect of O/A ratio on percentages of metals extracted. $T$ = 25 ± 1 °C; pH = 5.3; $t_{eq}$ = 15 min. Initial aqueous concentrations [g $L^{-1}$]: Li: 2.70; Ni: 1.91; Co: 15.2.

Co was concentrated to over 100 g $L^{-1}$ by stripping loaded Cyanex 272 with 2 M $H_2SO_4$ at O/A = 10–50 (Figure 8). The Cyanex 272 solution for the stripping experiments (Figure 8) was prepared by single extraction at pH = 4.95 and O/A = 1.5. Stripping the loaded Cyanex 272 once at O/A = 10 backextracted all Li and Ni, and 99.9% Co, resulting in 99.8 wt.% relative purity for Co. Scrubbing of Li and Ni can be applied before stripping to obtain the Co fraction at an even higher purity [10,33].

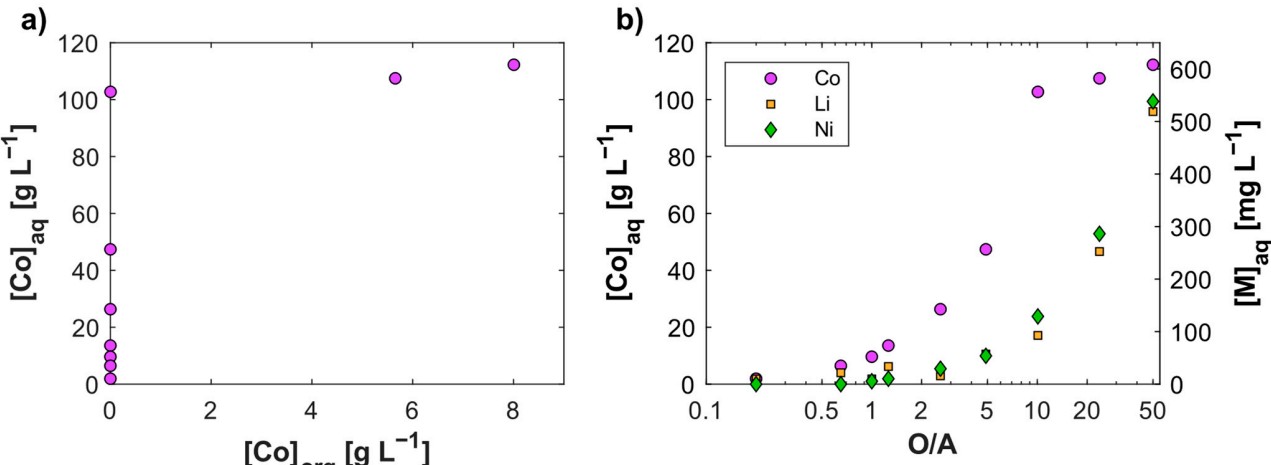

**Figure 8.** Backextraction of Co, Li, and Ni by 2 M $H_2SO_4$ from loaded 0.8 M Cyanex 272 (diluted in Exxsol D80): (**a**) Equilibrium distributions of Co; (**b**) Effect of O/A ratio on the metal concentrations in the stripping raffinate. $T$ = 25 ± 1 °C; $t_{eq}$ = 15 min. Initial concentrations in the organic phase: 10.2 g $L^{-1}$ Co, 10.5 mg $L^{-1}$ Li, and 10.1 mg $L^{-1}$ Ni. Note the logarithmic O/A-axis and secondary *y*-axis for Ni and Li in subfigure (**b**).

A total of 99.8% Co was extracted by 0.8 M Cyanex 272 in three countercurrent stages operating at S/F = 2.5, leaving approximately 30 mg $L^{-1}$ Co in the raffinate (Table 3). The relative purity of Co in the extract was 99.2 wt.% (Table 3). A major benefit of the current process scheme is that complete removal of Co is not required when NCM synthesis mixtures are produced instead of pure Ni fractions because the remaining Co in the raffinate will eventually be extracted together with Ni by D2EHPA (Figure 1). Therefore, the production of NCM synthesis mixtures is more flexible than the production of purified Ni fractions. A solution containing 2 g $L^{-1}$ Ni allows approximately 0.25 g $L^{-1}$ Co to maintain the stoichiometric ratio of NCM811 and, respectively, 0.66 g $L^{-1}$ Co can be allowed to produce NCM622 sulfate mixture (see Section 3.5). The tolerable amount of Co in the CoSX raffinate also depends on the level of Co in the Mn-D2EHPA (Figure 1). Comparable removal rate (98.8%) and relative purity of Co in the organic phase (99.4 wt.%) was obtained in a single extraction at O/A = 2 (Figure 7), meaning that additional extraction stages do not bring significant advantages with the process concept in Figure 1.

**Table 3.** Compositions of the extract and raffinate in three-stage countercurrent extraction of Co from the MnSX raffinate by 0.8 M Cyanex 272. S/F = 2.5; $T = 21 \pm 1$ °C; $\tau_{mix}$ = 5.7 min. Concentrations in [mg $L^{-1}$]. Numbering of the stages is started from the organic feed stage.

| | **[Li]** | **[Na]** | **[Mn]** | **[Co]** | **[Ni]** | **pH$_{avg}$** |
|---|---|---|---|---|---|---|
| MnSX raffinate (aq. feed) | 2180 | 8610 | 111 | 16,200 | 1990 | Stage 1: 5.35 ± 0.43 |
| Raffinate | 1930 | 18,900 | 0 | 27.4 | 2390 | Stage 2: 4.86 ± 0.31 |
| Extract | 3.89 | 0 | 46.0 | 6490 | 4.14 | Stage 3: 3.68 ± 0.19 |

The unexpectedly high [Ni] in the raffinate (2.39 g $L^{-1}$; Table 3) was a temporary spike likely due to pH fluctuation. Once the pH stabilized, the excess Ni was backextracted and resulted in a higher analytical concentration. This might also explain the mass balance error in the Li concentrations. As to Mn and Co, the mass balance is closed because they are extracted at a significantly lower pH than Ni and Li. The small amount of Mn in the aqueous feed (0.11 g $L^{-1}$) had a negligible effect on the separation of Co from Li and Ni by 0.8 M Cyanex 272 because the extractant was not even near saturation (Table 3).

### 3.4. Separation of Ni and Li

Several extraction stages are required for nearly complete removal (≥99%) and pre-concentration of Ni by 0.8 M D2EHPA at pH = 4.7 (Figure 9a). A high percentage of Ni (98.1%) was extracted in two countercurrent stages operating at S/F = 0.5 and pH = 7.0–7.7 (Table 4) but $[Ni]_{org}/[Li]_{org}$ remained relatively low. If Ni is extracted from a solution that contains a small amount of Co, the remaining Co is also extracted during the extraction of Ni with both D2EHPA and Cyanex 272 (see Section 3.1). Lowering of the O/A ratio did not result in a decrease in $[Li]_{org}$ (Figure 9b), which suggests that $Ni^{2+}$ ions cannot displace $Li^+$ from D2EHPA. The co-extracted Li can likely be recovered (see Section 3.5), so the extraction of Li does not need to be eliminated but it is recommended to maximize the $[Ni]_{org}/[Li]_{org}$ ratio during the extraction of Ni (see Section 3.5). Any residual Ni, Co, or Mn in the Li-rich raffinate can be precipitated as, e.g., mixed hydroxides, which can be recycled to the process feed. Li can be precipitated from the Ni-, Co-, and Mn-free raffinate as $Li_2CO_3$.

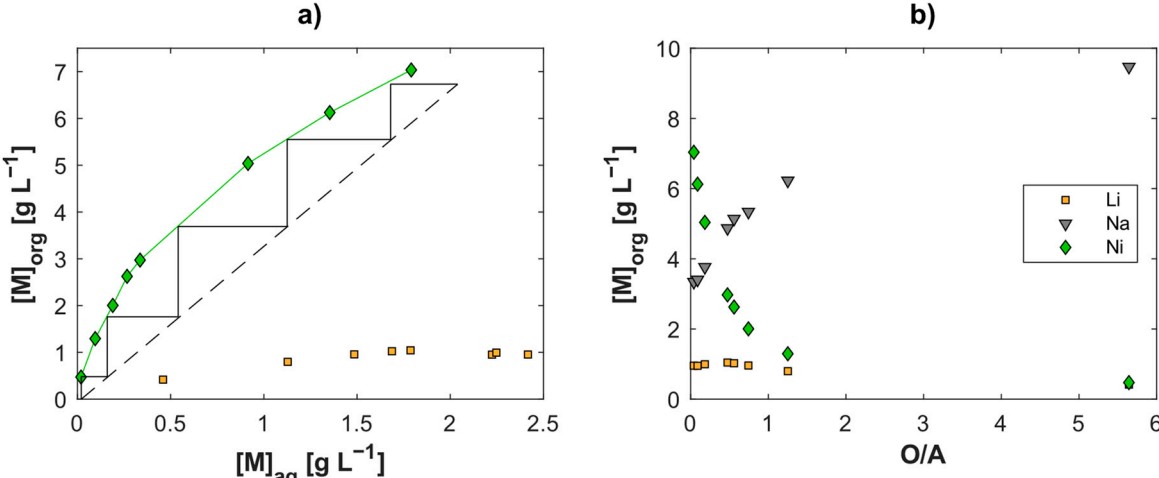

**Figure 9.** (**a**) A McCabe–Thiele analysis applied on the equilibrium distribution of Ni between 0.8 M D2EHPA (modified by 5 vol-% TBP and diluted in Exxsol D80) and a sulfate solution containing initially 2.41 g $L^{-1}$ Li, 2.04 g $L^{-1}$ Ni, and 20.8 g $L^{-1}$ Na; and (**b**) metal concentrations in the organic phase at different O/A ratios. $T = 25 \pm 1$ °C; pH = 4.7; $t_{eq}$ = 15 min. The operating line was calculated with S/F = 0.3 and 99% Ni removal.

**Table 4.** Compositions of the extract and raffinate in two-stage countercurrent extraction of Ni and Li from the CoSX raffinate by pre-neutralized 0.8 M D2EHPA. S/F = 0.5; $T = 21 \pm 1$ °C; $\tau_{mix}$ = 4 min. Concentrations in [mg $L^{-1}$]. Pre-neutralization: 81.5 mL 10 M NaOH in one liter of 0.8 M D2EHPA (modified by 5 vol-% TBP). Numbering of the stages is started from the organic feed stage.

|  | **[Li]** | **[Na]** | **[Mn]** | **[Co]** | **[Ni]** | **pH$_{avg}$** |
|---|---|---|---|---|---|---|
| CoSX raffinate (aq. feed) | 2580 | 20,900 | 0 | 98.5 | 2000 | Stage 1: 7.19 ± 0.12 |
| Raffinate | 1680 | 26,700 | 0 | 1.03 | 38.7 | Stage 2: 7.66 ± 0.10 |
| Extract | 1770 | 9150 | 0 | 191 | 3830 |  |

*3.5. Production of the NCM Synthesis Mixture*

Mixing the MnSX extract (Table 2) with the NiSX extract (Table 4) at 0.15:1 volumetric ratio yields NCM-D2EHPA where $n$(Ni):$n$(Mn) = 8.06 and $n$(Ni):$n$(Co) = 14.16. The Ni:Co:Mn stoichiometric ratio can be adjusted accurately to, for example, 8:1:1 by changing the mixing ratio of the D2EHPA extracts as well as through the addition of the concentrated CoSO$_4$ raffinate (Figure 8a) after complete stripping of the NCM-D2EHPA. However, the NCM-D2EHPA for the continuous stripping experiment (Table 5) was, for practical reasons, loaded with metals in a single contact instead of mixing two extracts from separate countercurrent extractions. Therefore, the initial [Na]$_{org}$ was unnecessarily high (Figure 9b).

**Table 5.** Compositions of the extract and raffinate in continuous stripping of Ni, Co, Mn, Li, and Na from 0.8 M NCM-D2EHPA with 3 M H$_2$SO$_4$. S/F = 10; $T = 21 \pm 1$ °C; $\tau_{mix}$ = 7.3 min. Concentrations in [mmol $L^{-1}$].

|  | **[Li]** | **[Na]** | **[Mn]** | **[Co]** | **[Ni]** | **pH$_{avg}$** |
|---|---|---|---|---|---|---|
| NCM-D2EHPA (org. feed) | 55.2 | 391 | 11.2 | 11.1 | 89.9 |  |
| Raffinate | 228 | 1770 | 30.1 | 49.2 | 399 | 1.62 ± 0.08 |
| Extract | 9.38 | 40.4 | 6.68 | 1.55 | 11.8 |  |

Stripping of Ni, Co, and Mn from 0.8 M NCM-D2EHPA was incomplete with a substoichiometric amount of H$_2$SO$_4$, and the concentrations of Ni, Co, and Mn in the organic phase were not decreasing significantly until most ($\approx$75%) of the Na was stripped

(Figure 10). Table 5 and Figure 11 show the raffinate composition in continuous stripping of NCM811-D2EHPA. The total concentration of Ni, Co, and Mn ([NCM]) was increased from 112 mmol $L^{-1}$ to 478 mmol $L^{-1}$. Significantly higher [NCM] can be obtained from stripping of NCM-D2EHPA with optimized composition, sufficiently high stripping acid concentration, and S/F ratio (see Sections 3.2 and 3.3). In addition to the possibility of concentrating the NCM mixture to the level required for direct precursor precipitation, controlling the amount of Li and Na in NCM-D2EHPA is important because their back-extraction consumes the stripping acid and they have the lowest solubilities—as grams of metal per kg of $H_2O$—among the metal sulfates studied here (Figure 12). It is likely that Li can be recovered from the NCM synthesis mixture by carbonate precipitation after precipitating the NCM precursor because the solubility limit of LiOH in water enables 36 g Li per kilogram of water at 25 °C [34].

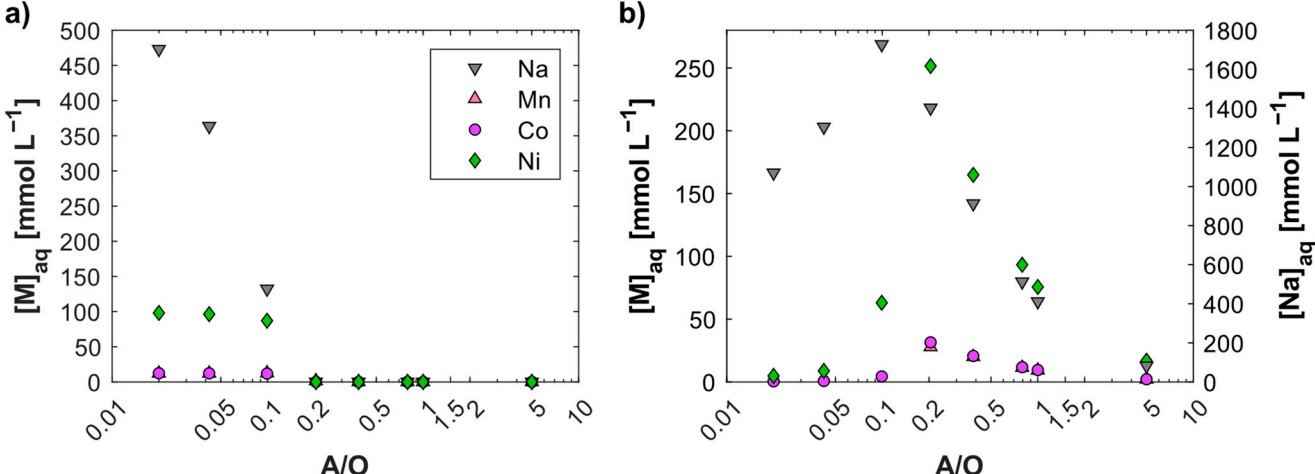

**Figure 10.** Equilibrium concentrations of metals in (**a**) the organic phase and (**b**) the aqueous phase in stripping of Ni, Co, Mn, and Na from 0.8 M NCM-D2EHPA with 2 M $H_2SO_4$. $T = 25 \pm 1$ °C; $t_{eq} = 15$ min. $[Ni]_{0,org} = 93.1$ mmol $L^{-1}$, $[Co]_{0,org} = 11.7$ mmol $L^{-1}$, $[Mn]_{0,org} = 11.5$ mmol $L^{-1}$, $[Na]_{0,org} = 529$ mmol $L^{-1}$.

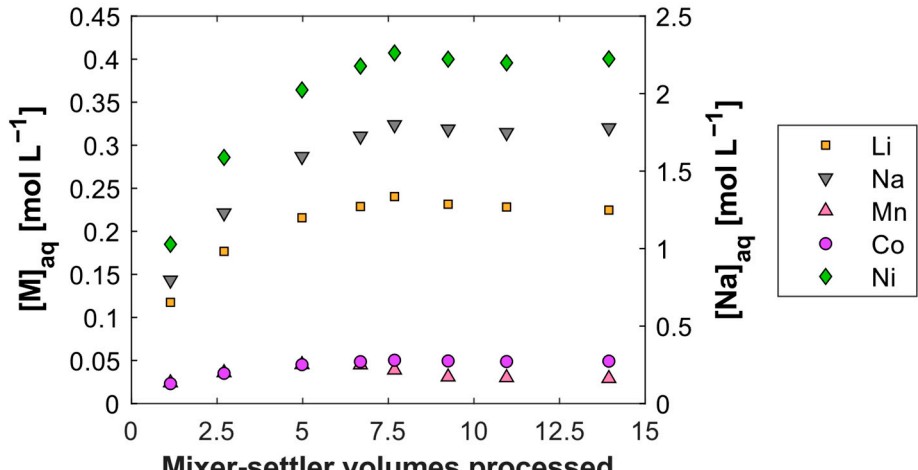

**Figure 11.** Metal concentrations in the raffinate during continuous stripping of 0.8 M NCM-D2EHPA with 3 M $H_2SO_4$. $T = 21 \pm 1$ °C, S/F = 10; $\tau_{mix} = 7.3$ min. $[Li]_{0,org} = 55.2$ mmol $L^{-1}$, $[Ni]_{0,org} = 89.9$ mmol $L^{-1}$, $[Co]_{0,org} = 11.1$ mmol $L^{-1}$, $[Mn]_{0,org} = 11.2$ mmol $L^{-1}$, $[Na]_{0,org} = 391$ mmol $L^{-1}$.

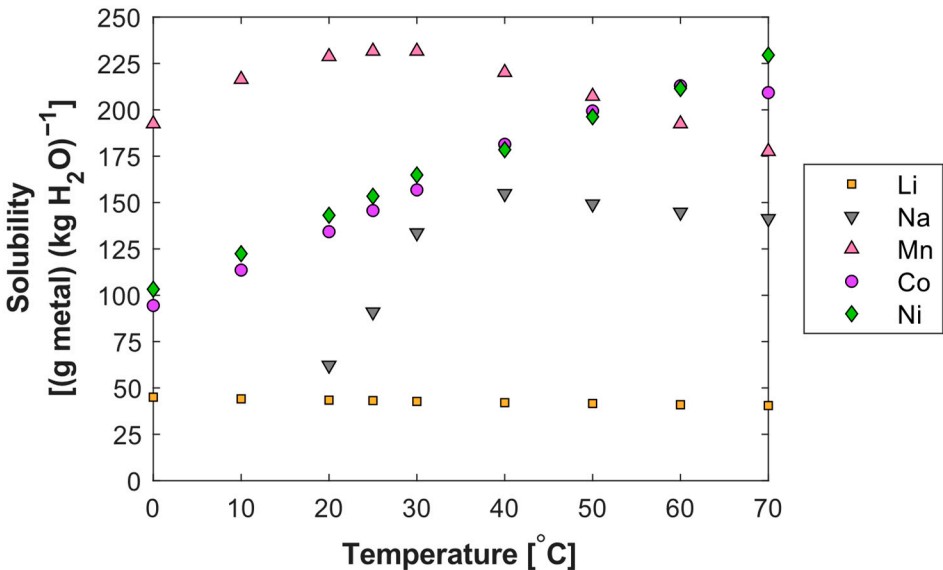

**Figure 12.** Solubilities of the metal sulfates in water (binary systems) at different temperatures given as grams of metal cations per kilogram of water. Data from [34,35].

The mass balance in Table 5 is not closed because the extract-to-raffinate volumetric flow ratio differed from S/F = 10. The change in volumetric phase ratio can easily be observed visually, especially in batch extraction. The changes in phase volumes are likely explained by the break-up of microemulsion or reversed micelle structures and by the release of hydrated water, and this must be considered in process optimization.

Preparing a Ni-rich sulfate solution from a Co-rich LIB leachate without additional input of transition metal salts is a task with increased complexity (Figure 1). When the molar ratios of Ni, Co, and Mn in the LIB leachate match with a commercial NCM cathode composition or when additional $NiSO_4$, $CoSO_4$, and $MnSO_4$ are used to adjust the solution composition [12–14], a more straightforward solvent extraction process than the concept in Figure 1 can be used. Adjusting the Ni:Co:Mn molar ratio in the Co-rich leachate (Table 1) to 8:1:1 using transition metal salts would require approximately 590 kg $NiSO_4·6H_2O$ and 42 kg $MnSO_4·H_2O$ per 1.0 $m^3$ of leachate. The requirement for compensating the deficit is high since much more metals should be introduced into the process than is delivered into it from the 1.0 $m^3$ of leachate. Then, the solution should be diluted if 2 mol $L^{-1}$ is the target total concentration, and 1.4–1.5 $m^3$ of NCM811 synthesis mixture would be obtained. The process concept in Figure 1 produces approximately 21 L of NCM811 synthesis mixture (at 2 mol $L^{-1}$ total concentration of Ni, Co, and Mn) from 1 $m^3$ of leachate (Table 1), assuming 100% material recovery. The material loops of the transition metals are closed in the concept of Figure 1. In addition, purified $Li_2SO_4$, $CoSO_4$ and $MnSO_4$ fractions are obtained.

## 4. Conclusions

In this study, a solvent extraction process concept was developed and demonstrated experimentally for the processing of Co-rich LIB leachates into Ni-rich solutions that are suitable for producing, for example, NCM811 or NCM622 cathode precursors. Furthermore, it is possible to obtain Li and excess Mn and Co in high purity into their own fractions. The proposed process offers an alternative to producing NCM cathode precursors from pure $NiSO_4$, $CoSO_4$, and $MnSO_4$ salts. The process is robust because total separation of the metals is not required, and the compositions of the streams can be adjusted by internal recycling.

94.2% Mn was extracted by 0.8 M D2EHPA from the simulated LIB leachate in three countercurrent stages operating at pH 2.1–2.5, $T$ = 21 ± 1 °C, and S/F = 0.8. Part of the Mn-rich D2EHPA, which also contains Co, Li, and Ni, can be blended with Ni-rich D2EHPA from another stage of the process to form NCM-D2EHPA. The NCM-D2EHPA is stripped

using moderately concentrated $H_2SO_4$ at a high S/F ratio to obtain the NCM mixture, which can be used in the precursor synthesis. Co, Li, and Ni are scrubbed from the excess Mn-rich D2EHPA to enable the production of purified $MnSO_4$.

All of the remaining Mn and 99.8% Co was extracted by 0.8 M Cyanex 272 in three countercurrent stages operating at pH 3.7–5.4, $T$ = 21 ± 1 °C, and S/F = 2.5. The relative purity of Co in the extract was over 99 wt.%. Subsequently, 98.1% Ni was extracted from the Co-barren raffinate by pre-neutralized 0.8 M D2EHPA in two countercurrent stages operating at pH 7.2–7.7, $T$ = 21 ± 1 °C, and S/F = 0.5. The raffinate from continuous Ni extraction contained 1.68 g $L^{-1}$ Li, 26.7 g $L^{-1}$ Na, 1 mg $L^{-1}$ Co, and 38.7 mg $L^{-1}$ Ni. Continuous stripping of the NCM-D2EHPA increased the total concentration of Ni, Co, and Mn from 112 mmol $L^{-1}$ to 478 mmol $L^{-1}$. The different steps of the process were not optimized.

**Author Contributions:** Conceptualization, N.J., S.V. and T.S.; data curation, N.J. and S.V.; formal analysis, N.J. and S.V.; funding acquisition, S.V. and T.S.; investigation, N.J.; methodology, N.J., S.V. and T.S.; project administration, S.V. and T.S.; resources, S.V. and T.S.; supervision, S.V. and T.S.; validation, N.J. and S.V.; visualization, N.J.; writing—original draft, N.J.; writing—review and editing, S.V. and T.S. All authors have read and agreed to the published version of the manuscript.

**Funding:** The research was funded by the BATCircle2.0 project (main funder Business Finland, grant number 44412/31/2020).

**Institutional Review Board Statement:** Not applicable.

**Informed Consent Statement:** Not applicable.

**Data Availability Statement:** The data presented in this study are available within the article.

**Acknowledgments:** The experimental assistance from Antti Tolvanen (LUT, Finland) and Siiri Närvänen (LUT, Finland) is gratefully acknowledged.

**Conflicts of Interest:** The authors declare no conflict of interest. The funders had no role in the design of the study; in the collection, analyses, or interpretation of data; in the writing of the manuscript; or in the decision to publish the results.

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
