# Peer review of "Direct Production of Ni–Co–Mn Mixtures for Cathode Precursors from Cobalt-Rich Lithium-Ion Battery Leachates by Solvent Extraction"

_metals, doi:10.3390/met12091445_

Round 1

Reviewer 1 Report

1. Compared with the traditional precursor fabrication process, what are the advantages of the work done in this paper.

2. The horizontal and vertical coordinates in the figure need to be adjusted and optimized to make the data points clearer. (Line 315, 325)

3. Can the agents mentioned in this article such as (Cyanex 272 and D2EHPA) be recycled?

4. There are two errors in the citations in the text, please modify them. (Line 187, 320)

5. Please adjust the citation format strictly according to the format of this journal.

Reviewer 2 Report

I appreciate the work that has been put together in this manuscript. While I do not consider the approach to be completely novel, there is significant value on the presented work on the field of process engineering, and the simplification of downstream separation process to achieve the obtention of cathode grade materials. The authors did a good job explaining a complex flow diagram through the document.

I would suggest maybe include some labels in Figure 1 to guide that the dotted/dashed lines are organic streams and the continuous lines are aqueous.

There are some references that did not show the right citation format.
